# Fluoride Treatment and In Vitro Corrosion Behavior of Mg-Nd-Y-Zn-Zr Alloys Type

**DOI:** 10.3390/ma15020566

**Published:** 2022-01-12

**Authors:** Pham Hong Quan, Iulian Antoniac, Florin Miculescu, Aurora Antoniac, Veronica Manescu (Păltânea), Alina Robu, Ana-Iulia Bița, Marian Miculescu, Adriana Saceleanu, Alin Dănuț Bodog, Vicentiu Saceleanu

**Affiliations:** 1Faculty of Material Science and Engineering, University Politehnica of Bucharest, 313 Splaiul Independentei, District 6, 060042 Bucharest, Romania; hongquan83fpt@gmail.com (P.H.Q.); or antoniac.iulian@gmail.com (I.A.); florin.miculescu@upb.ro (F.M.); veronica.paltanea@upb.ro (V.M.); alina.ivanov@upb.ro (A.R.); ana.iulia.bita@upb.ro (A.-I.B.); 2Academy of Romania Scientist, 54 Splaiul Independentei, 050094 Bucharest, Romania; 3Faculty of Electrical Engineering, University Politehnica of Bucharest, 313 Splaiul Independentei, District 6, 060042 Bucharest, Romania; 4Faculty of Medicine, Lucian Blaga University of Sibiu, 2A Lucian Blaga Street, 550169 Sibiu, Romania; adriana.saceleanu@ulbsibiu.ro (A.S.); vicentiu.saceleanu@gmail.com (V.S.); 5Faculty of Medicine and Pharmacy, University of Oradea, 10 P-ta 1 December Street, 410073 Oradea, Romania; alinbodog@gmail.com

**Keywords:** Mg-Nd-Y-Zn-Zr alloys, fluoride treatment, in vitro testing, corrosion behavior

## Abstract

Fluoride conversion coatings on Mg present many advantages, among which one can find the reduction of the corrosion rate under “in vivo” or “in vitro” conditions and the promotion of the calcium phosphate deposition. Moreover, the fluoride ions released from MgF_2_ do not present cytotoxic effects and inhibit the biofilm formation, and thus these treated alloys are very suitable for cardiovascular stents and biodegradable orthopedic implants. In this paper, the biodegradation behavior of four new magnesium biodegradable alloys that have been developed in the laboratory conditions, before and after surface modifications by fluoride conversion (and sandblasting) coatings, are analyzed. We performed structural and surface analysis (XRD, SEM, contact angle) before and after applying different surface treatments. Furthermore, we studied the electrochemical behavior and biodegradation of all experimental samples after immersion test performed in NaCl solution. For a better evaluation, we also used LM and SEM for evaluation of the corroded samples after immersion test. The results showed an improved corrosion resistance for HF treated alloy in the NaCl solution. The chemical composition, uniformity, thickness and stability of the layers generated on the surface of the alloys significantly influence their corrosion behavior. Our study reveals that HF treatment is a beneficial way to improve the biofunctional properties required for the studied magnesium alloys to be used as biomaterials for manufacturing the orthopedic implants.

## 1. Introduction

Today, most of the orthopedic implants used in traumatology are made from Ti-based and stainless steels, because they exhibit excellent mechanical properties and are tolerated by the human body until the fracture is healing [1]. In order to manufacture a high-quality implant, the biocompatibility of the material and the matching of mechanical properties to bone must be taken into account. The main side effect of the metallic implants consist of metal ion emission due to corrosion phenomenon and inflammation at the implant site, or dangerous tissue reaction as cell apoptosis or necrosis can occur [1,2,3,4,5]. Another important limitation of the classical metallic implants for orthopedic surgery is the so-called “stress shielding” phenomenon, which is due to the metal high value of Young’s modulus in comparison with that of human bone. Various failure analysis studies about the failed metallic implants used in orthopedic surgery reveal these problems [6,7,8].

In order address these aspects, the researchers were focused, at the beginning, to the solutions such as different biocompatible coatings applied on the metallic materials surface [3,4,5,6,9,10,11] or to create different composite materials for orthopedic applications [7,8]. However, a new paradigm that appears in the last decade looks to be more suitable to solve some problems of the classic metallic implants used in orthopedic surgery. This paradigm is represented by the biodegradable metals like magnesium alloys, to manufacture performant implants [12,13,14].

Magnesium (Mg) is a unique material, which presents physical and mechanical biocompatibility to human bone. Its Young’s modulus and density have similar values to those of bone [2,9,15]. Mg bone implants could be classified as bone fixation device (bone screws, pins and plates) and bone tissue engineering. In different scientific papers, no important difference between Ti and Mg screws was observed after 6 months post-surgery [16,17,18,19,20,21,22,23]. The main drawback of the innovative Mg implants is their higher corrosion rate in physiological media. Due to the increased value of this factor, a high degradation rate of the implant mechanical properties can be noticed, followed by a premature failure of the device, which takes place before the patient tissues’ healing. In Figure 1, the optimal degradation response of Mg alloy implant versus bone fracture healing is presented.

Hydrogen release is another potential disadvantage, and it is the result of a cathodic reaction, which can seriously affect the healing procedure. Corrosion of the magnesium alloy has as result a surface oxide/hydroxide layer formation that is not stable in the chloride ions presence, because these types of ions convert the hydroxide layer into a very soluble compound the so-called magnesium chloride. The hydrogen gas evolution is linked to gas bubbles, which are subcutaneously located or placed in the implant neighborhood [24,25]. This can generate a separation process between tissue layers [26]. Some in vitro medical studies have determined that the critical tolerance level of hydrogen must be lower than 0.01 mL/cm^2^/day [27] and they assume that the hydrogen evolution does not have important interference with the body healing process, but the magnesium corrosion rate must be carefully controlled in the first two weeks after the patient surgery, to be in the biological range. Usually, Mg alloys are characterized by a low value of the mechanical strength in conjunction with their degradation rate. Some methods were proposed in literature as alloying the Mg with biocompatible elements or grain refinement, to increase the mechanical properties [28]. Due to this fact Mg based implants are used in unload-bearing position, and they are also characterized by inadequate fatigue properties [29]. One of the problems identified in the literature consist of the fact that there are a few scientific papers that treat the functional testing of the implants and the time interval is not always clearly identified, in which the implant loses its mechanical stability due to corrosion phenomenon.

Other methods for improving the mechanical properties and adapting the corrosion process to the clinical needs for biodegradable Mg alloys are thermo-mechanical processing of the surface [29,30,31,32,33,34,35,36].

In Figure 2 a schematically representation of the surface modifications, which can be applied for Mg alloys, is presented.

**Fluoride conversion coatings on Mg** presents a lot of advantages as it follows: they reduce the corrosion rate of Mg for analysis made in “in vivo” or “in vitro” conditions, due to their property to control the pH increase and hydrogen gas accumulation; they are characterized by a good biocompatibility; they sustain a gradual Mg degradability and offer a very good cellular response for MG63 and MC3T3-E1 mouse cells; they promote the calcium phosphate deposition; the fluoride ions released from MgF_2_ did not present cytotoxic effects and inhibit the biofilm formation; they are not harmful to blood cells; they are very suitable for cardiovascular stents and biodegradable orthopedic implants. Different approaches for preparation of fluoride conversion coatings are presented in the scientific literature [37,38,39].

It is assumed that the MgF_2_ layer is formed on the surface after the reaction of magnesium with HF solution (Figure 3), as shown in Equation (1). A dynamic equilibrium should be established between the dissolution of magnesium and the deposition of MgF_2_ [40,41,42].
(1)$${\mathrm{Mg}}_{\left(s\right)}+2{\mathrm{HF}}_{\left(\mathrm{aq}\right)}\to {\;\mathrm{MgF}}_{2\left(s\right)}+{\;H}_{2\left(g\right)}$$

Because the layer of MgF_2_ deposited on the surface of the magnesium alloy is insoluble, it functions as a gradually increasing barrier. Over time, the dissolution of magnesium should be reduced and thus MgF_2_ deposition could be slowed down. Depending on the concentration of the HF solution, an oxidation reaction takes place during the formation of magnesium fluoride as follows:(2)$${\mathrm{Mg}}_{\left(s\right)}+2H_{2}O_{\left(l\right)}\to \;\mathrm{Mg}{\left(\mathrm{OH}\right)}_{2\left(s\right)}+{\;H}_{2\left(g\right)}$$

Because Mg(OH)_2_ layer is not stable in acid solution the following reactions should occur [43]:(3)$$\mathrm{Mg}{\left(\mathrm{OH}\right)}_{2\left(s\right)}+2{\mathrm{HF}}_{\left(\mathrm{aq}\right)}\;\to {\mathrm{MgF}}_{2\left(s\right)}+2H_{2}O_{\left(l\right)},$$
(4)$$\mathrm{Mg}{\left(\mathrm{OH}\right)}_{2\left(s\right)}\to {\mathrm{MgO}}_{\left(s\right)}+2H_{2}O_{\left(l\right)},$$

The lower the concentration of hydrofluoric acid, the higher the amount of hydroxides and oxides formed on the surface of the magnesium alloys [43].

Most studied Mg alloys contain Al and other alloying elements, and commercial Mg alloys are well researched. Some Mg alloys, such as AZ31, AM60B and WE43 are attempted as biomaterials. Al, Mn, Zr and RE elements are employed to improve the mechanical and corrosion properties. However, some of these elements could have negative effects on the human bone [44,45,46].

The purpose of this work is to study the in vitro corrosion behavior of some Mg-Nd-Y-Zn-Zr alloys type before and after a combination of surface modification: sandblasting and/or fluoride treatment. The novelty of this manuscript is related not just to the corrosion evaluation of the novel biodegradable magnesium alloys type Mg-Nd-Y-Zn-Zr, but also to the clarification on the effect of sandblasting before fluoride treatment in the case of these biodegradable magnesium alloys.

## 2. Materials and Methods

### 2.1. Sample Preparation

Ingots of MRI 201s and MRI 202s magnesium alloys with the composition presented in Table 1 were supplied by the DSM Company (Beersheba, Israel). The alloys were cut into pieces of 15 × 15 × 5 mm, then mechanically polished up to 1200 grit with silicon carbide abrasive papers (SiC). After that all the substrates were rinsed with acetone, ethanol and distilled water successively.

Three groups of samples from each type of alloy were obtained: samples from the first group were treated with HF solution, those from the second group were sandblasted with alumina particles, and those from the third group were first sandblasted and then treated with HF solution. The sandblasting of the samples was made using alumina (Al_2_O_3_) with a particle size of 200 µm (purchased from Poka, Romania). The jet was placed perpendicularly to the sample surface (the sandblasting angle was set to 90°) at a distance of 10 mm with a pressure of 2 MPa. The sandblasting time was set to 20 s. The fluoride layer was obtained after immersing of the alloy samples in an aqueous solution of hydrofluoric acid of 40 wt.% (purchased from Sigma-Aldrich, Darmstadt, Germany) at room temperature for a time of 24 h. At the end the samples were rinsed with distilled water and dried in the air [37,39,40].

The experimental samples were coded according to the details shown in Table 2.

### 2.2. Materials Characterization—Structural and Surface Analysis

The surface morphology and elemental composition of the samples before and after immersion testing were examined under a scanning electron microscope (SEM, Philips XL 30 ESEM TMP) coupled with energy X-ray dispersive spectroscopy system (EDS, EDAX Sapphire Spectrometer). The structural aspects of the experimental samples were determined by using a Panalytical X-Pert PRO Diffractometer (Malvern, UK), which works in Bragg-Brentano symmetric geometry. The contact angle measurements (CA) were made in order to determinate the wettability properties of the surface that is very important in the case of implantable biomaterials. A KRÜSS DSA30 Drop Shape Analysis System was used, and obtained images were processed by aligning the tangent at the profile of the sessile drop at the point of contact with the surface. Contact angle measurements were made in triplicate and an average value was calculated.

### 2.3. Electrochemical Tests

Corrosion resistance was investigated using polarization resistance technique at 37 ± 0.5 °C in NaCl (Sodium chloride purchased from Sigma-Aldrich, Darmstadt, Germany) using a PARSTAT 4000 Potentiostat/Galvanostat (Princeton Applied Research, Oak Ridge, TN, USA). An area of 1 cm^2^ was exposed to the electrolyte in experiments. A typical three-electrode electrochemical cell with the following set-up: sample as working electrode (WE), a platinum electrode was used as a counter electrode (CE), and saturated calomel (SCE) as reference electrode (RE), was used for all electrochemical measurements. The open circuit potential (OCP) was monitored for 1 h, starting right after the sample’s immersion in the electrolyte and the Tafel plots were recorded from ±200 mV (vs. OCP) at a scanning rate of 1 mV/s. All measurements were achieved according to the ASTM G5-14e1 standard [47].

### 2.4. Immersion Test

The test consisted in immersion of the experimental samples, at 37 ± 0.5 °C, in the polyethylene containers with 50 cm^3^ of sodium chloride with a pH value of 7.0. The corrosion behavior of the experimental samples was observed by weighing them after removing from NaCl solution and subsequent drying. The experimental samples were weighed with a balance type WLC 2/A2 produced by RADWAG Balances and Scales, at the beginning and after 1, 3, 5, 7 and 14 days, at the same hour. The NaCl solution was changed every day. The weight loss was calculated with the following formula:(5)$$\Delta m\left[\%\right]\;=\frac{M_{i}-M_{f}}{M_{i}}*100,$$
where M_i_ is the initial mass value recorded at the beginning of the experiment; M_f_ is the final mass value at the end of the experiment.

The surface morphology of the experimental samples after immersion testing and the composition of the corrosion products were analyzed by scanning electron microscopy with an energy dispersive spectroscopy system.

## 3. Results and Discussions

### 3.1. Materials Characterization—Structural and Surface Analysis

#### 3.1.1. Scanning Electron Microscopy—Energy Dispersive X-ray Spectroscopy

The SEM images obtained after the surface analysis of the untreated, fluoride treated samples, and sandblasted samples, are shown in Figure 4.

For original magnesium alloys MRI201s and MRI202s, SEM images show the precipitation of a secondary phase at the grain boundary. A more uniform distribution of the secondary phase at the grain boundary is observed in the case of MRI202s alloy.

It is observed that the treatment of both experimental alloy samples (MRI201s and MRI202s) with HF conduct to a similar smoother surface without changing the morphology of the substrate. The layer of MgF_2_ formed at the surface is very thin, that permit a clear visualization of the grain boundaries.

In the case of sandblasted samples, SEM image exhibit for both alloys many deep cavities over the whole surface. Obviously, the roughness of the samples surface increased with the sandblasting process. As a result of this fact, the contact area with the corrosion medium increases leading to the decrease of the corrosion resistance of these samples compared to the samples treated with HF.

#### 3.1.2. XRD

The XRD spectra obtained on all experimental magnesium alloys samples are shown in Figure 5. Compared to the original magnesium alloys MRI201s and MRI202s which revealed the presence of magnesium and MgO phases in HF-treated alloys (MRI201s-H, MRI201s-SH, MRI202s-H, MRI202s-SH) the presence of MgF_2_ in the conversion layer is observed.

In XRD patterns for MRI201s-S, MRI201s-SH, MRI202s-S and MRI202s-SH samples Al_2_O_3_ phase were detected, indicating residual particles after the sandblasting treatment. In all XRD patterns for treated magnesium alloys the magnesium phase is observed, phase from the substrate which is present because layer formed at the surface is very thin.

#### 3.1.3. Contact Angle

The obtained results are presented in Table 3 and the corresponding images in Figure 6.

As can be seen from the Figure 6 the contact angle decreases in the case of both samples with HF treated surface, from 61° to 20° in case of the MRI201s alloy and from 55° to 16° in the case of MRI202s alloy.

Results obtained for sandblasted samples (MRI201s-S and MRI202s-S) showed that the hydrophobicity have increased and lost their hydrophilic character.

### 3.2. Electrochemical Corrosion Behavior

Figure 7 and Figure 8 show the open circuit potential curves and Tafel plots of magnesium alloys in NaCl solution.

Based on Tafel extrapolation the main electrochemical parameters were extracted: open circuit potential (E_OC_); corrosion potential (E_corr_); corrosion current density (i_corr_); cathodic Tafel slope (β_c_); anodic Tafel slope (β_a_).

In the Table 4 are presented the main electrochemical parameters of corrosion processes.

The polarization resistance (Rp) was calculated by using the Stern–Geary equation:(6)$$R_{p}=\frac{1}{2.3\cdot i_{\mathrm{corr}}}\cdot \frac{\beta_{a}\cdot \left|\beta_{c}\right|}{\beta_{a}+\left|\beta_{c}\right|},$$
where R_p_ is the polarization resistance; β_a_ and β_c_ are the anodic and cathodic Tafel slops respectively, and i_corr_ is the corrosion current density.

The corrosion rate calculation was performed according to ASTM G102-89 (2015) [48] using the following formula:(7)$$\mathrm{CR}={\;K}_{i}*\frac{i_{\mathrm{corr}}}{\rho }*\mathrm{EW},$$

According to the obtained results presented in Table 4 it can be observed that after sandblasted all alloys have a poor corrosion resistance and the treatment with HF have enhanced the electrochemical behavior of the investigated magnesium alloys samples regardless of surface treatment.

The electrochemical results reveal that after chemical treatment with HF solution of both investigated magnesium alloys, they exhibit the lowest corrosion current density, highest polarization resistance and a corrosion resistance of almost 10 times higher for MRI201s-H alloy, and almost four times higher for MRI202s-H alloy compared with untreated magnesium alloys. In relation to untreated alloys, for the MRI201s-S and MRI202s-S samples, the electrochemical results revealed the degradation of the surface layer and a decrease of the corrosion resistance caused by the initial sandblasting of the samples. Furthermore, after the immersion test the degradation rates for both sandblasted samples are significantly increased mainly due to the high surface roughness and deep cavities formed at the surfaces, which increase substrate activity.

### 3.3. Immersion Tests

The evolution of the degradation rate assessed by determining weight loss is presented in Figure 9. In the case of both experimental alloys (MRI201s and MRI202s), the treatment with HF induces a reduction of the degradation process even the samples were sandblasted previously or not.

The degradation rate of untreated MRI201s and MRI202s alloys is higher due to the existence of a high concentration of Cl^-^ ions in the test medium. Corrosion of MRI201s and MRI202s magnesium alloys is a redox process of magnesium oxidation, a process coupled with the reduction and formation of hydrogen both from hydrogen ions and/or the hydrogen atom in the water molecule [49,50]. The Cl¯ ions present in the test environment transform the Mg(OH)_2_ layer formed at the surface (according to the Equation (2)) into MgCl_2_, a soluble precipitate in solution [51,52]. This process causes an increase in hydroxide ions (OH¯) nearby the surface of the sample leading to an increase in the pH of the solution. The corrosion process takes place until the corrosion layer (Mg(OH)_2_) that forms in these regions, reaches a saturation point, at a pH value of at least 10.4.

The weight loss of MRI201s-H and MRI202s-H samples (HF treated samples) and MRI201s-SH and MRI202s-SH samples (sandblasted with HF-treated samples) was lower than in the case of untreated samples for the entire stage. This is due to the MgF_2_ layer formed on the surface of the samples which acts as an inhibitor and protects them from corrosion.

The results obtained for sandblasted samples (MRI201s-S and MRI202s-S) indicate a significant increase in weight loss compared to the original samples ((MRI201s and MRI202s). In general, the alloys degradation rate increased with a higher surface roughness [53,54,55]. Sandblasting of the magnesium alloys surface induces a high dislocation density and the formation of deep cavities, which increase substrate activity.

The surface morphologies of the experimental samples after 14 days of immersion in sodium chloride solution are shown in Figure 10.

By analyzing the two initial Mg alloys, it can be seen that the MRI 202 alloy is more degraded. This is probably due to the higher amount of Y in the chemical composition of the MRI 202 alloy, which gives a higher homogeneity of the grains in the microstructure. The difference between the behavior of the two alloys is maintained in the case of samples treated with HF, but is equalized after the application of the blasting treatment or after blasting and treatment with HF.

Regarding the surface modification treatments applied on the initial alloys, it is obvious that the surface sandblasting treatment leads to a more accentuated degradation of the sandblasted samples, regardless of the composition of the alloy. At the same time, the surface has a rougher topography and favors the biomaterial-tissue interaction. This positive effect induced by blasting is also maintained in the case of experimental samples that were treated with HF after blasting, but slightly attenuated.

Macroscopic investigations of the surface of the experimental samples after the immersion test demonstrate that the experimental samples sandblasted and subsequently treated with HF have the best surface properties in terms of biomaterial-tissue interaction.

In the SEM images on MRI201s, investigated after 5 days of immersion (Figure 11), it can be observed the formation at the surface of dense spherical corrosion products. After immersion in NaCl for 14 days, a layer of corrosion products with a large number of clusters appeared on the samples surfaces. The presence of this layer at 14 days of immersion suggests that the material exchange process is fast but without the formation of cracks on the samples surface that would produce an aggressive corrosion process. For the MRI202s sample the process is more intense the obtaining of a dense layer of corrosion products is visible after 7 days of immersion in NaCl solution.

The relatively smooth surface of the HF-treated samples suggests that the chemical conversion layer of magnesium fluoride is denser than that of the untreated and sandblasted samples, which explain the better corrosion resistance. After 5 days of immersion in NaCl the presence of corrosion cracks is not observed on the surface of the HF-treated samples, while in the case of sandblasted samples relatively small cracks appear which indicates a more intense corrosion process. The sandblasted samples surfaces become uneven with coarse cluster corrosion products. The formation of corrosion cracks is observed also in the case of sandblasted with HF-treated samples at 7 days of immersion in NaCl solution.

## 4. Discussions

The results revealed that the untreated sample suffered considerably from the environment while the fluoride treated sample showed significantly better performance for longer exposure time in corrosive solution. Surface treatment of MRI201s and MRI202s alloys by immersion in HF solution ensures a uniform, smoother surface without changing the morphology of the substrate, which results in a better protection of the substrate. We also investigate the effect of sandblasting treatment of MRI201s and MRI202s alloys using particle of alumina on in vitro degradation behavior.

Some studies demonstrate that in the case of corrosion testing by electrochemical methods or immersion tests of some biodegradable magnesium alloys after fluoride treatment, the HF concentration is important, because a more uniform, dense and thick coating layer was obtained when a higher HF concentration (40%) was used [36,37,40]. Electrochemical tests prove that fluoride layer determines an increase of the corrosion resistance, but several corrosion dots are present. Ren et al. [56] studied calcium phosphate glass/MgF_2_ double layered composite coating on AZ31 magnesium alloy. It was concluded after microstructural analysis and classical corrosion tests that the corrosion current density is reduced at 0.6μA/cm^2^ and the charge transfer resistance increases with two magnitude order. The pH of the double coated sample decreases, and a large adhesion strength was observed. Immersion tests prove that the CaP glass/MgF_2_ composite coating provides a high corrosion protection for the Mg substrate, and it is very suitable for using for implants, because it is biodegradable. Reza Bakhsheshi-Rad et al. [57] has applied a fluoride treatment in the case of a Mg-Ca binary alloy, but on a Mg-0.5Ca alloy and conclude that Mg-0.5Ca treated with 40% HF presents a low degradation kinetics and a high biocompatibility.

An important role in the functionality and biocompatibility of implantable devices has also the surface wettability, property quantified by determining the contact angle values. On the surfaces with high roughness, the water drop will remain in its spherical shape and will not adhere to the surface. It is also known that a more hydrophilic surface, with a smaller contact angle, reflects good adhesiveness, good wettability, and higher solid surface free energy. Thus, if we talk about osseointegration, a small contact angle, so a hydrophilic surface, will improve cell adhesion while a hydrophobic surface can affect cell adhesion and protein denaturation, leading to the rejection of implantable material. Chengyu Xu et al. [58] showed that surface induce different kinds of cell responses. Fibroblasts prefer smoother surfaces, epithelial cells attached only to the smoothest surfaces while osteoblast cells prefer rougher surface.

In terms of corrosion, according to Baboian [59] and Mansfeld [60], a material is resistant to the corrosive attack of a media when all the following are met: a more electropositive corrosion potential (E_corr_), a low corrosion current density (i_corr_) and a higher polarization resistance (R_p_). After corrosion experiments two samples stands out, recording the smallest i_corr_ and higher R_p_. Those samples are MRI201s-H (i_corr_ = 4.203 µA/cm^2^ and Rp = 8.997 kΩ ·cm^2^) and MRI202s-H (i_corr_ = 14.87 µA/cm^2^ and Rp = 0.727 kΩ ·cm^2^).

As a general remark after the corrosion experiments it can be highlight that:For both investigated magnesium alloys the sandblasting treatment leads to a worsening of corrosion resistance.After chemical treatment with HF of initial surfaces, both investigated magnesium alloys exhibit the lowest corrosion current density and highest polarization resistance.After the chemical treatment with HF of sandblasted surfaces the corrosion resistance is highly improved for both magnesium alloys.

The formation of MgF_2_ layer on the magnesium alloys and sandblasted magnesium alloys surfaces has led to a decrease in corrosion rate, the best values being obtained for MRI201s alloy (MRI201s-H, CR = 0.094 mm/y and MRI201s-SH, CR = 0.342 mm/y). The uniformity, thickness, porosity, chemical composition, and stability of the MgF_2_ layer generate significant influence of the corrosion behavior of the experimental samples.

## 5. Conclusions

The results showed an improved corrosion resistance for HF treated MRI201s and MRI202s alloy in the NaCl solution. SEM images after 14 days of immersion in NaCl exhibits the formation at samples surface of a corrosion products layer with a large number of clusters in case of MRI201s and MRI202s samples; a dense layer of MgCl_2_ conversion products for MRI201s-H and MRI202s-H samples and corrosion cracks for MRI201s-SH, MRI201s-S, MRI202s-SH and MRI202s-S samples. The chemical composition, uniformity, thickness and stability of the layers generated on the surface of the alloys significantly influence their corrosion behavior.

We can conclude that, for both magnesium alloys investigated, HF treatment of magnesium alloy surfaces reduces the corrosion process, while sandblasting is not an efficient method of surface modification to achieve this goal. The chemical composition of investigated magnesium alloys does not influence significantly the degradation process; regardless of the type of treatment applied the evolution is similar. In the case of all samples, a more accelerated increase in weight loss is observed in the first 5 days, after which the intensity of the degradation process decreases. That means that after the first few days, a more stable Mg(OH)_2_ layer will be formed at the surface.

In conclusion, our study reveals that HF treatment is a beneficial way to improve the biofunctional properties required for magnesium alloys type MRI201s and MRI202s to be used as biomaterials for manufacturing the orthopedic implants. For obtaining a better interaction between the magnesium alloys and surrounding tissue, we propose sandblasting followed by HF treatment as a biodegradable magnesium alloy processing protocol that allow obtaining optimal surface properties.

## Figures and Tables

**Figure 1 materials-15-00566-f001:**
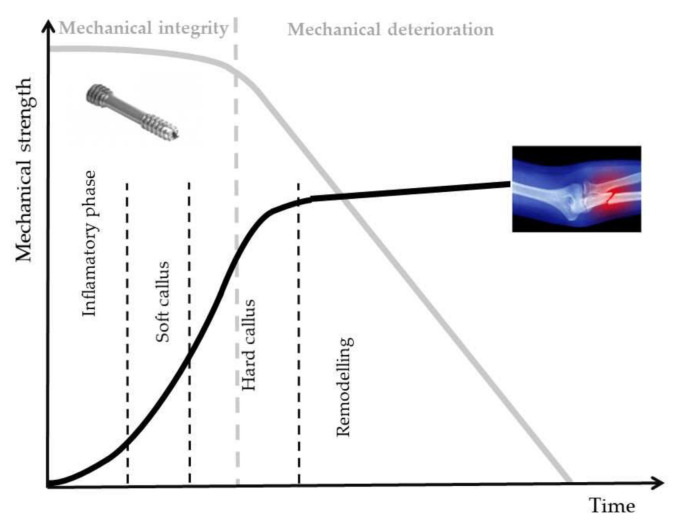
Degradation behavior of Mg alloy implants in bone fracture healing made in optimal conditions, adapted from [17].

**Figure 2 materials-15-00566-f002:**
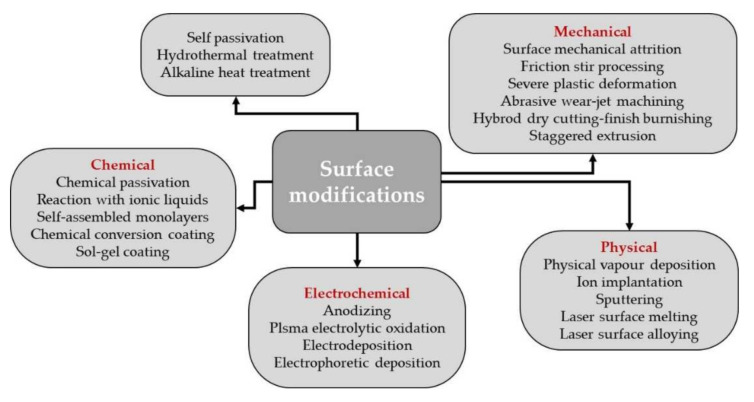
Schematically representation of the surface modifications that can be applied in the case of Mg based alloys.

**Figure 3 materials-15-00566-f003:**
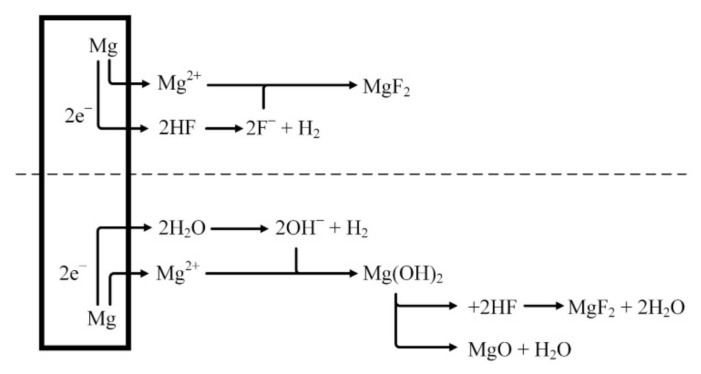
Schematical representation of fluoride conversion coatings on magnesium.

**Figure 4 materials-15-00566-f004:**
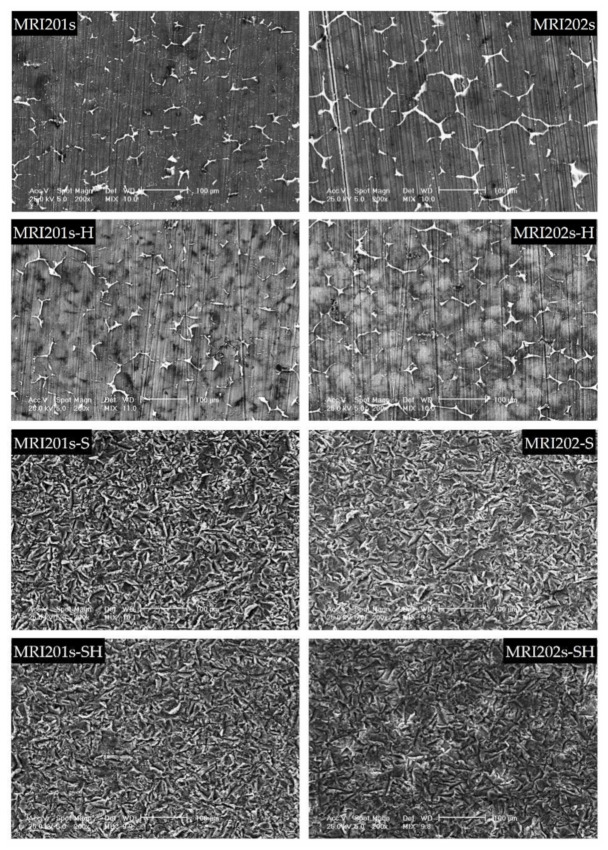
SEM images obtained after the surface analysis for untreated (MRI201s and MRI202s), fluoride treated samples (MRI201s-H and MRI202s-H), sandblasted samples (MRI201s-S and MRI202s-S) and combined treated samples (MRI201s-SH and MRI202s-SH)—scalebars are 100 microns.

**Figure 5 materials-15-00566-f005:**
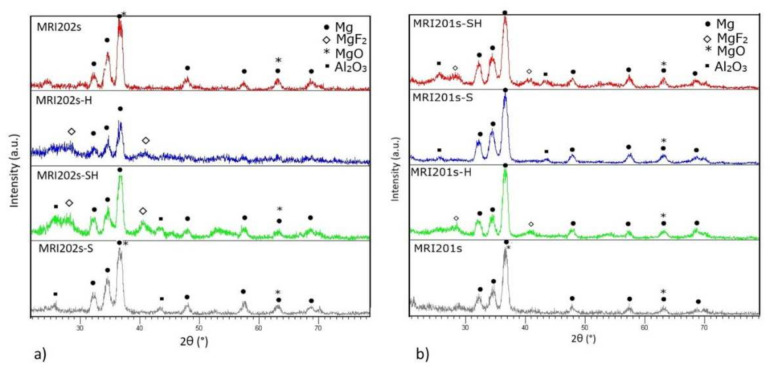
XRD diffraction patterns on untreated and treated alloys (**a**) MRI202s and (**b**) MRI201s.

**Figure 6 materials-15-00566-f006:**
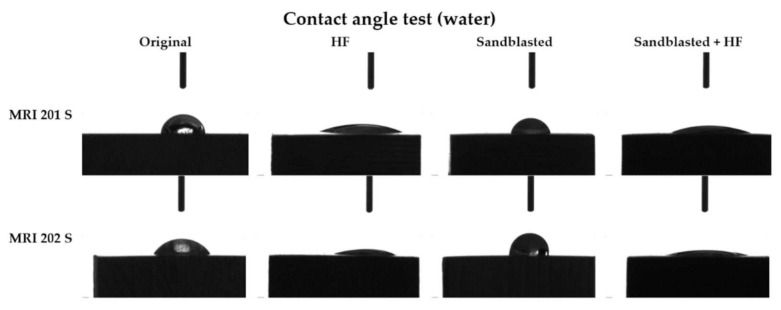
The contact angles values for each type of samples.

**Figure 7 materials-15-00566-f007:**
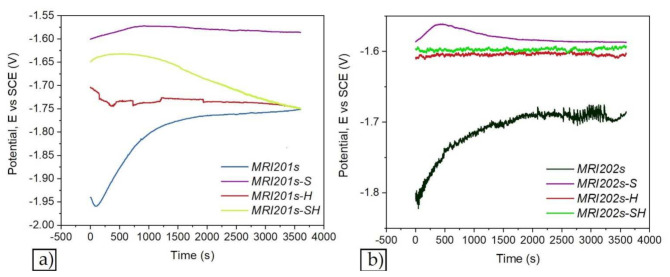
The open circuit potential curves of magnesium alloys: MRI201s alloy (**a**), MRI202s alloy (**b**).

**Figure 8 materials-15-00566-f008:**
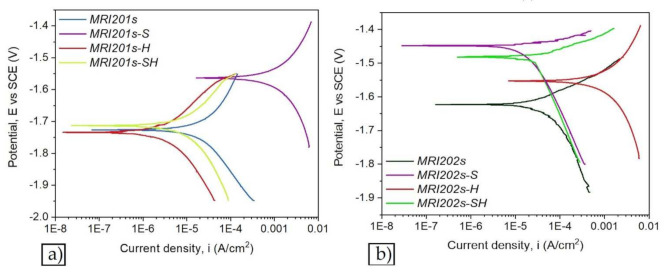
The Tafel plots of magnesium alloys: MRI201s alloy (**a**), MRI202s alloy (**b**).

**Figure 9 materials-15-00566-f009:**
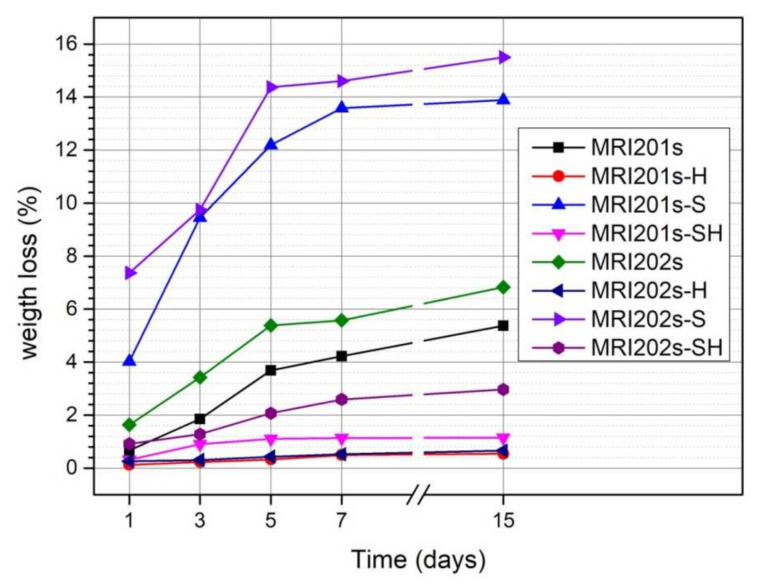
Weight loss of untreated and treated MRI201s and MRI202s samples after 1, 3, 5, 7 and 14 days of immersion in NaCl solution.

**Figure 10 materials-15-00566-f010:**
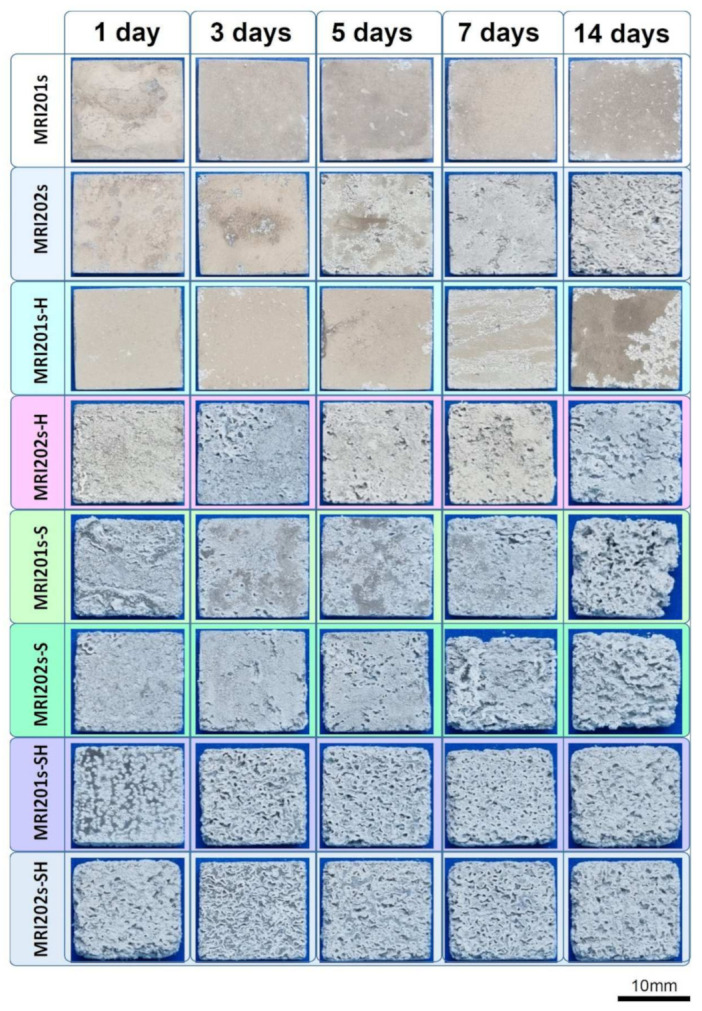
Surface morphologies of the experimental samples after 1, 3, 5, 7 and 14 days of immersion in sodium chloride solution.

**Figure 11 materials-15-00566-f011:**
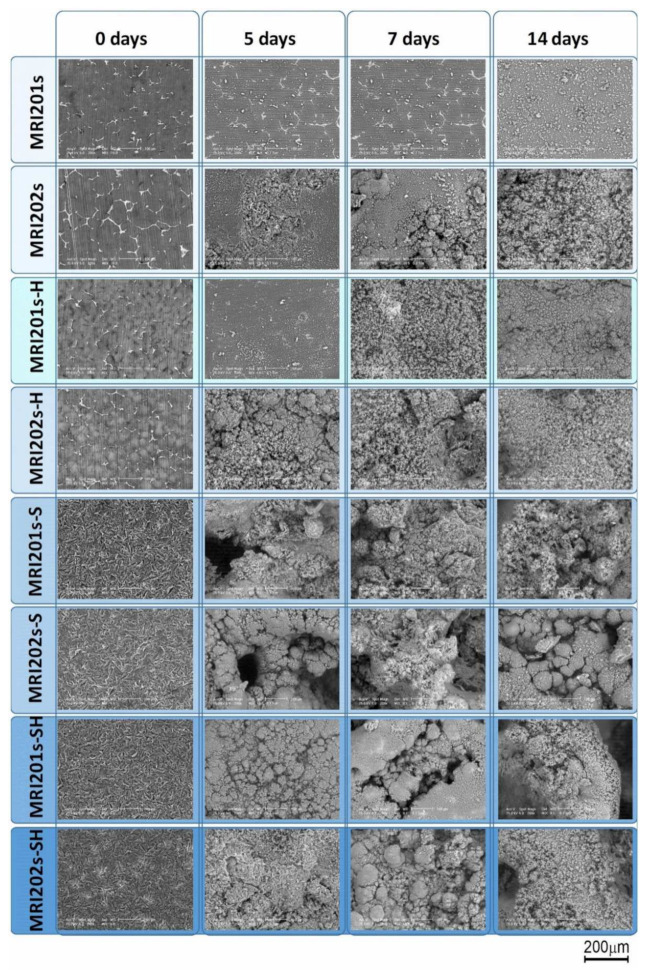
Surface morphologies of the experimental samples after 0, 5, 7 and 14 days of immersion in sodium chloride solution.

**Table 1 materials-15-00566-t001:** The chemical compositions of the experimental Mg alloys (wt%).

Samples	Zn (%)	Zr (%)	Y (%)	Nd (%)	Mg (%)
MRI201s	0.3	0.6	2.10	3.2	Bal.
MRI202s	0.3	0.4	0.21	3.1	Bal.

**Table 2 materials-15-00566-t002:** Coding of experimental samples (scalebars from images are 5 mm long).

Samples	Macro Image	Treatment Applied
MRI201s	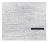	original MRI 201s alloy
MRI201s-H	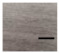	MRI 201s alloy treated with HF
MRI201s-S	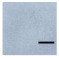	Sandblasted MRI 201s alloy
MRI201s-SH	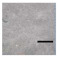	Sandblasted MRI 201s alloy treated with HF
MRI202s	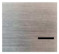	original MRI 202s alloy
MRI202s-H	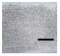	MRI 202s alloy treated with HF
MRI202s-S	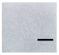	Sandblasted MRI 202s alloy
MRI202s-SH	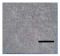	Sandblasted MRI 202s alloy treated with HF

**Table 3 materials-15-00566-t003:** Contact angle measurements values (degrees).

Sample	Contact Angle Values (Degrees)
MRI201s	61 ± 0.757
MRI201s-H	21 ± 0.897
MRI201s-S	78 ± 0.741
MRI201s-SH	18 ± 0.729
MRI202s	55 ± 0.987
MRI202s-H	23 ± 0.331
MRI202s-S	80 ± 0.699
MRI202s-SH	19 ± 0.509

**Table 4 materials-15-00566-t004:** Main obtained electrochemical parameters.

No.	Sample	E_oc_ (mV)	E_corr_ (mV)	i_corr_ (µA/cm^2^)	β_c_ (mV)	β_a_ (mV)	Rp (kΩxcm^2^)	CR (mm/y)
1.	MRI201s	−1750	−1726	38.926	266.77	280.00	1.525	0.874
2.	MRI201s-H	−1749	−1733	4.203	202.01	152.73	8.997	0.094
3.	MRI201s-S	−1585	−1562	7943	2318	922.20	0.036	178.446
4.	MRI201s-SH	−1750	−1712	15.249	290.90	202.28	3.401	0.342
5.	MRI202s	−1685	−1622	55.703	268.93	87.48	0.515	1.257
6.	MRI202s-H	−1603	−1447	14.87	244.67	27.68	0.727	0.335
7.	MRI202s-S	−1587	−1552	4280	938.74	477.46	0.032	96.637
8.	MRI202s-SH	−1593	−1481	30.354	332.79	45.68	0.575	0.685

## Data Availability

Not applicable.

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
