# Peer review of "Fluoride Treatment and In Vitro Corrosion Behavior of Mg-Nd-Y-Zn-Zr Alloys Type"

_materials, 2022, doi:10.3390/ma15020566_

Round 1
Reviewer 1 Report
Pham Hong Quan et al studied fluoride treatment and in vitro corrosion behavior of Mg-Nd-Y-Zn-Zr alloys. The work is interesting and can be considered for acceptance after minor modifications.
1. It is recommended that the author set the colors of the curves in Figure 8 and 9 to be easily distinguishable.
2. The resolution of Figure 4 needs to be improved further. Please set clear scale bar for Figure 4, 11 and 12.
3. It is suggested that the authors carry out further characterization of the alloy with the best corrosion resistance. It is suggested that the author provide sectional images, high-magnification SEM images and mappings of different elements toward the alloy after corrosion. This can enrich the variety of data and increase the persuasive power of data.
4. The authors may consider adding XPS data, which is helpful to determine the surface products. This one is optional.
5. Some references are too old, it is suggested to cite the latest achievements in recent three years. Moreover, references with formatting errors need to be checked carefully to meet the format requirements of the journal. The following literature is for reference.
[1] Microstructure and property modifications in surface layers of a Mg-4Sm-2Al-0.5Mn alloy induced by pulsed electron beam treatments, JOURNAL OF MAGNESIUM AND ALLOYS 2021, 9(1), 216-224.
[2] Recent progress in superhydrophobic and superamphiphobic coatings for magnesium and its alloys, JOURNAL OF MAGNESIUM AND ALLOYS 2021, 9(3), 748-778.
[3] Stearic Acid Coated MgO Nanoplate Arrays as Effective Hydrophobic Films for Improving Corrosion Resistance of Mg-Based Metallic Glasses, Nanomaterials 2020, 10, 947; doi:10.3390/nano10050947.
Author Response
Thank you for analysing our manuscript.
- It is recommended that the author set the colors of the curves in Figure 8 and 9 to be easily distinguishable.
The colours were changed in the figures 8 and 9 for a better distinguish.
- The resolution of Figure 4 needs to be improved further. Please set clear scale bar for Figure 4, 11 and 12.
The resolution was changed in figure 4 and scale bars were added in figures 4 , 11 and 12
- It is suggested that the authors carry out further characterization of the alloy with the best corrosion resistance. It is suggested that the author provide sectional images, high-magnification SEM images and mappings of different elements toward the alloy after corrosion. This can enrich the variety of data and increase the persuasive power of data.
We thank the reviewer for the suggestion. The section analysis of the samples will be presented together with the results of in vivo tests and are part of the topic of the next article. Also Figure 4 shows the SEM images at the most representative magnification, so that the details needed for comparison are best highlighted. For this analysis, images from 25x to 2000x were obtained, those presented in the figure being the ones that optimally highlight the comparison of the effect pursued in the article. If images were shown at higher magnifications, secondary precipitates would be visible only in the case of half of the samples, which would not lead to better information.
- The authors may consider adding XPS data, which is helpful to determine the surface products. This one is optional.
Thank you for the comment. Since the XPS analysis provides information only from the surface, with depths of the order of nanometers, we considered that the analysis of EDS, which allows obtaining quantitative data from depths of the order of nanometers is more relevant. According to the authors, an XPS analysis would not add value to this work, as the thickness of the corrosion layers is much higher than the tangible level by this method.
- Some references are too old, it is suggested to cite the latest achievements in recent three years. Moreover, references with formatting errors need to be checked carefully to meet the format requirements of the journal. The following literature is for reference.
[1] Microstructure and property modifications in surface layers of a Mg-4Sm-2Al-0.5Mn alloy induced by pulsed electron beam treatments, JOURNAL OF MAGNESIUM AND ALLOYS 2021, 9(1), 216-224.
[2] Recent progress in superhydrophobic and superamphiphobic coatings for magnesium and its alloys, JOURNAL OF MAGNESIUM AND ALLOYS 2021, 9(3), 748-778.
[3] Stearic Acid Coated MgO Nanoplate Arrays as Effective Hydrophobic Films for Improving Corrosion Resistance of Mg-Based Metallic Glasses, Nanomaterials 2020, 10, 947; doi:10.3390/nano10050947.
We thank the reviewer for the suggestions. We carefully analysed the suggested references and we found relevant dates in two references shown below that became references 45 and 46.
[45] Microstructure and property modifications in surface layers of a Mg-4Sm-2Al-0.5Mn alloy induced by pulsed electron beam treatments, JOURNAL OF MAGNESIUM AND ALLOYS 2021, 9(1), 216-224.
[46] Recent progress in superhydrophobic and superamphiphobic coatings for magnesium and its alloys, JOURNAL OF MAGNESIUM AND ALLOYS 2021, 9(3), 748-778.
Reviewer 2 Report
- The reference for Figure 1 to be mentioned.
- Better place figure 3 after the explanation of the figure.
- How the material treatment process (sand blasting and HF treatment) parameters are considered? Give at least reference proof.
- In the SEM image (Figure 4), no secondary precipitates are visible in the selected scale of image. Moreover, surface roughness or deep cavities evidence are not seen in the SEM image? Please give description on concentrating the details reflected by the image.
- Please nomenclature the images (Figure 4) about the details obtained under different material treatments. Also, add more explanations for the Figure 4.
- Please compare the figures 5 and 6 (XRD diffraction patterns) for the possible inference.
- Contact angles for MRI201s-H and MRI202s-H (Table 3) is same, whereas in figure 7 result, seems to be different. Why? Please explain.
- Some discussion and critical analysis is required for the section “2. Electrochemical corrosion behaviour” and result (Figures 8 and 9), because the work is depend fully on Electrochemical corrosion behaviour. Please include.
- The details in the SEM images (Figures11 and 12) are to be marked. More in-depth discussion is required as the outcome of this result.
- Table 3 data analysis, it is mentioned as sand blast specimen has poor corrosion resistance. It is the fact, because sand blasting is not only increases surface area but also increases strain energy in the surface and subsurface layers by strain hardening. In sand blast specimen, the explanation may be around corrosion effect due to surface roughness obtained and strain hardening index improvement happened.
- In the Conclusion, the effect of metal treatment, type of phase present and contact angle on the corrosion behaviour is to be included.
- In Reference 24, 25, 33, 34, 42, 44, 45, 46, 47, 48, 49 and 51 to 55, in the title of the paper, only the first letter of the first word is of upper-case whereas, in the remaining references, all words of the first letter is of upper-case. Please correct them for the format uniformity.

Author Response
- The reference for Figure 1 to be mentioned.
Thank you for the observation. The following text was inserted: Adapted from reference [17].
- Better place figure 3 after the explanation of the figure.
We thank the reviewer for the comment. We placed the figure so that the submitted manuscript to fill as much as possible the page (its position within the paper will be discussed with the editorial board of the journal), but we agree that was inappropriately placed, and we will do our best to solve this issue.
- How the material treatment process (sand blasting and HF treatment) parameters are considered? Give at least reference proof.
The details are given between the line 142 and 152. References used was [37, 39, 40]
- In the SEM image (Figure 4), no secondary precipitates are visible in the selected scale of image. Moreover, surface roughness or deep cavities evidence are not seen in the SEM image? Please give description on concentrating the details reflected by the image.
The main purpose for the image 4 was to put in evidence the different topography of each type of samples. Also, the surface roughness is clearly put in evidence for the sandblasted samples. Figure 4 shows the SEM images at the most representative magnification, so that the details needed for comparison are best highlighted. For this analysis, images from 25x to 2000x were obtained, those presented in the figure being the ones that optimally highlight the comparison of the effect pursued in the article. If images were shown at higher magnifications, secondary precipitates would be visible only in the case of half of the samples, which would not lead to better information. We just mention the presence of the secondary visible phase at the grain boundary without a complex analysis of this due to the focus of the current article and the relevance for the current research.
- Please nomenclature the images (Figure 4) about the details obtained under different material treatments. Also, add more explanations for the Figure 4.
We consider that is not necessary to complicatedly load the images from figure 4 with punctual details that cand be easily observed by the general readers. We usually make this explanatory images for the manuscripts focused on microstructural characterization or for analysis of the corrosion product formed at the surface in the case of biodegradable magnesium alloys but is not the case for this manuscript.
- Please compare the figures 5 and 6 (XRD diffraction patterns) for the possible inference.
We combined the XRD images for a better understanding. The resulted image is Figure 5
- Contact angles for MRI201s-H and MRI202s-H (Table 3) is same, whereas in figure 7 result, seems to be different. Why? Please explain.
We thank the reviewer for the observation. At the contact angle in table 3 there was a typing error and the mistyped one value was changes. For the sample MRI202s-H, the correct value is 23±0.331
- Some discussion and critical analysis is required for the section “2. Electrochemical corrosion behaviour” and result (Figures 8 and 9), because the work is depend fully on Electrochemical corrosion behaviour. Please include.
Thank you for the observation. We included the following discussion:
The electrochemical results reveal that after chemical treatment with HF solution of both investigated magnesium alloys exhibit the lowest corrosion current density, highest polarization resistance and a corrosion resistance of almost 10 times higher for MRI201s-H alloy, respectively almost 4 times higher for MRI202s-H alloy compared with untreated magnesium alloys. In relation to untreated alloys, for the MRI201s-S and MRI202s-S samples, the electrochemical results revealed the degradation of the surface layer and a decrease of the corrosion resistance caused by the initial sandblasting of the samples. Also, after the immersion test the degradation rates for both sandblasted sam-ples are significantly increased mainly due to the high surface roughness and deep cavi-ties formed at the surfaces, which increase substrate activity.
- The details in the SEM images (Figures11 and 12) are to be marked. More in-depth discussion is required as the outcome of this result.
The following discussion is inserted: By analysing the two initial Mg alloys, it can be seen that the MRI 202 alloy is more degraded. This is probably due to the higher amount of Y in the chemical composition of the MRI 202 alloy, which gives a higher homogeneity of the grains in the microstructure. The difference between the behaviour of the two alloys is maintained in the case of samples treated with HF, but is equalized after the application of the blasting treatment or after blasting and treatment with HF. Regarding the surface modification treatments applied on the initial alloys, it is obvious that the surface sandblasting treatment leads to a more accentuated degradation of the sandblasted samples, regardless of the composition of the alloy. At the same time, the surface has a rougher topography and favours the biomaterial-tissue interaction. This positive effect induced by blasting is also maintained in the case of experimental samples that were treated with HF after blasting, but slightly attenuated. Macroscopic investigations of the surface of the experimental samples after the immersion test demonstrate that the experimental samples sandblasted and subsequently treated with HF have the best surface properties in terms of biomaterial-tissue interaction. In the SEM images on MRI201s, investigated after 5 days of immersion (Figure 12), it can be observed the formation at the surface of dense spherical corrosion products. After immersion in NaCl for 14 days, a layer of corrosion products with a large number of clusters appeared on the samples surfaces. The presence of this layer at 14 days of immersion suggests that the material exchange process is fast but without the formation of cracks on the samples surface that would produce an aggressive corrosion process. For the MRI202s sample the process is more intense the obtaining of a dense layer of corrosion products is visible after 7 days of immersion in NaCl. The relatively smooth surface of the HF-treated samples suggests that the chemical conversion layer of magnesium fluoride is denser than that of the untreated and sand-blasted samples, which explain the better corrosion resistance. After 5 days of immersion in NaCl the presence of corrosion cracks is not observed on the surface of the HF-treated samples, while in the case of sandblasted samples relatively small cracks appear which indicates a more intense corrosion process. The sandblasted samples surfaces become uneven with coarse cluster corrosion products. The formation of corrosion cracks is observed also in the case of sandblasted with HF-treated samples at 7 days of immersion in NaCl solution.
- Table 3 data analysis, it is mentioned as sand blast specimen has poor corrosion resistance. It is the fact, because sand blasting is not only increases surface area but also increases strain energy in the surface and subsurface layers by strain hardening. In sand blast specimen, the explanation may be around corrosion effect due to surface roughness obtained and strain hardening index improvement happened.
According to the short time used for immersion test and our previous studies on other biodegradable magnesium alloys, we consider that the strain hardening of the substrate do not influence the corrosion resistance. As is visible in figure 11, the samples were not degraded in profound way and didn’t loss a higher amount of material. Anyway, we appreciate your comment, and we try to realize new research in order to follow the hardness of different section after sandblasting process and their influence on the long term biodegradation rate.
- In the Conclusion, the effect of metal treatment, type of phase present and contact angle on the corrosion behaviour is to be included.
Thank you for the suggestion. We add these comments in conclusion section: We can conclude that, for both magnesium alloys investigated, HF treatment of magnesium alloy surfaces reduces the corrosion process, while sandblasting is not an efficient method of surface modification to achieve this goal. The chemical composition of investigated magnesium alloys does not influence significantly the degradation process; regardless of the type of treatment applied the evolution is similar. In the case of all samples, a more accelerated increase in weight loss is observed in the first 5 days, after which the intensity of the degradation process decreases. That means that after first days, a more stable Mg(OH)2 layer will be formed at the surface.
- In Reference 24, 25, 33, 34, 42, 44, 45, 46, 47, 48, 49 and 51 to 55, in the title of the paper, only the first letter of the first word is of upper-case whereas, in the remaining references, all words of the first letter is of upper-case. Please correct them for the format uniformity.
Thank you very much for the observation. The references were formatted accordingly.
Reviewer 3 Report
The authors present in paper: Fluoride treatment and in vitro corrosion behavior of Mg-Nd-Y-Zn-Zr alloys type, few very interesting results on commercial biodegradable Mg-base alloys.
Few minor aspects can be taken in consideration in order to improve the paper quality:
- a general English language check can be done
- L28: better explain the purpose of the immersion tests
- L29: replace simulated body fluid with NaCl solution
- L96+L108: a reference for each figure 2 and 3 is necessary , or the schematics are made by the authors ?
- L140: mention the type of percentage : wt or at ?, number of determination and standard deviation or give a reference of the material standard
- L154: insert scales (mm or cm) for macro-images of the surfaces
- L200: improve the scale quality - al least for the last 4 images of the figure
- L232: insert the contact angle measure
- L260: give a reference
- L297: insert scales in figure 11
- L308: after 0,5,7 and 14
- L309: Figure 12 : scales are not visible
- L317: NaCl solution
- L389: better structure the conclusions
Author Response
- a general English language check can be done
The English language was corrected
- L28: better explain the purpose of the immersion tests
Many thanks for the comment. The following explanation were added regarding the immersion tests at line 28.
We performed structural and surface analysis (XRD, SEM, contact angle) before and after applying different surface treatments. Also, we studied the electrochemical behaviour and biodegradation of all experimental samples after immersion test performed in NaCl solution. For a better evaluation, we also used LM and SEM for evaluation of the corroded samples after immersion test.
- L29: replace simulated body fluid with NaCl solution
Thank you for the observation and we are sorry for the inserted error. The simulated body fluid was replaced to NaCl solution.
- L96+L108: a reference for each figure 2 and 3 is necessary, or the schematics are made by the authors ?
Even the figures looks similar to others from literature, our figures are different and made by the authors.
- L140: mention the type of percentage : wt or at ?, number of determination and standard deviation or give a reference of the material standard
Sorry for the omission. We added (wt%) in the table caption
- L154: insert scales (mm or cm) for macro-images of the surfaces
Scales were added in the macro-images
- L200: improve the scale quality - al least for the last 4 images of the figure
The scale bars were improved for a better view.
- L232: insert the contact angle measure
Thank you. We made this.
- L260: give a reference
Thank you. Reference [46] ASTM G102-89 (2015), was added
- L297: insert scales in figure 11
Scales were inserted within the image
- L308: after 0,5,7 and 14
Thank you for the observation. We changed the figure caption accordingly.
- L309: Figure 12: scales are not visible
Scales were inserted within the image
- L317: NaCl solution
Thank you for the observation. We changed the sentence by adding “solution”.
- L389: better structure the conclusions
The conclusion section was restructured.
Round 2
Reviewer 2 Report
All corrections are incorporated.